# CHARACTER GENERATION THROUGH SELF-SUPERVISED VECTORIZATION

## ABSTRACT

The prevalent approach in self-supervised image generation is to operate on pixel level representations. While this approach can produce high quality images, it cannot benefit from the simplicity and innate quality of vectorization. Here we present a drawing agent that operates on stroke-level representation of images. At each time step, the agent first assesses the current canvas and decides whether to stop or keep drawing. When a 'draw' decision is made, the agent outputs a program indicating the stroke to be drawn. As a result, it produces a final raster image by drawing the strokes on a canvas, using a minimal number of strokes and dynamically deciding when to stop. We train our agent through reinforcement learning on MNIST and Omniglot datasets for unconditional generation and parsing (reconstruction) tasks. We utilize our parsing agent for exemplar generation and type conditioned concept generation in Omniglot challenge without any further training. We present successful results on all three generation tasks and the parsing task. Crucially, we do not need any stroke-level or vector supervision; we only use raster images for training. Code will be made available upon acceptance.

## 1 INTRODUCTION

While, innately, humans sketch or write through strokes, this type of visual depiction is a more difficult task for machines. Image generation problems are typically addressed by raster-based algorithms. The introduction of generative adversarial networks (GAN) (Goodfellow et al., 2014), variational autoencoders (VAE) (Kingma & Welling, 2013) and autoregressive models (Van Oord et al., 2016) has led to a variety of applications. Style transfer (Gatys et al., 2015; Isola et al., 2017), photo realistic image generation (Brock et al., 2018; Karras et al., 2019), and super resolution (Ledig et al., 2017; Bin et al., 2017) are some of the significant instances of the advancing field. Additionally, Hierarchical Bayesian models formulated by deep neural networks are able to use the same generative model for multiple tasks such as classification, conditional and unconditional generation (Hewitt et al., 2018; Edwards & Storkey, 2016). These raster-based algorithms can produce high quality images, yet they cannot benefit from the leverage that higher level abstractions bring about.

Vector-level image representation intrinsically prevents models from generating blurry samples and allows for compositional image generation which eventually may contribute to our understanding of how humans create or replicate images (Lake et al., 2017). This idea, with the introduction of sketch-based datasets such as Omniglot (Lake et al., 2012), Sketchy (Sangkloy et al., 2016), and QuickDraw (Ha & Eck, 2017) has triggered a significant body of work in recent years. Stroke based image generation and parsing has been addressed with both vector supervised models and self-supervised generation. Of these, one prominent algorithm is Bayesian Program Learning (Lake et al., 2015), where a single model can be utilized for 5 tasks in the Omniglot challenge: (i) parsing, (ii) unconditional generation or, (iii) generating exemplars of a given concept, (iv) generating novel concepts of a type, and (v) one-shot classification. This approach is also shown to be scalable when supported by the representative capabilities of neural networks (Feinman & Lake, 2020b;a), however, it requires stroke-level or vector supervision, which is costly to obtain or simply non-existent. VAE/RNN (Ha & Eck, 2017; Cao et al., 2019; Chen et al., 2017; Aksan et al., 2020) and Transformer based models (Ribeiro et al., 2020; Lin et al., 2020) are other common methods applied to vector based image generation. Although impressive results have been presented, stroke-level supervision is required to train these models.

Recently, self-supervised (i.e. the absence of stroke-level supervision) stroke-based image generation has been addressed with Reinforcement Learning (RL) (Ganin et al., 2018; Mellor et al., 2019; Huang et al., 2019; Schaldenbrand & Oh, 2020). We call this approach self-supervised vectorization, since the vectorization of images is learned using only raster-images as supervision.

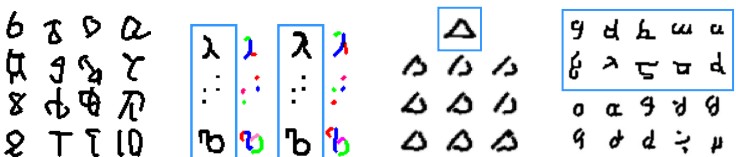

Figure 1: Our drawing agent can accomplish four different tasks. From left to right: it can generate novel characters, parse a given character into its strokes, generate new exemplars for a given character, and generate novel concepts (i.e. characters) given a type (i.e. alphabet). **Ours is the first stroke-based method to tackle all of the generation and parsing tasks in the Omniglot Challenge, without requiring any stroke-level supervision.**

These methods mostly focus on image reconstruction and their exploration in generation is limited. For example, none of them address the conditional generation problem, or, they need the number of actions (i.e. strokes) as input.

In this paper, we propose a self-supervised reinforcement learning approach where we train a drawing agent for character generation and parsing. Our drawing agent operates on the stroke-level (i.e. vector) representation of images. At each time step, our agent takes the current canvas as input and dynamically decides whether to continue drawing or stop. When a 'continue' decision is made, the agent outputs a program specifying the stroke to be drawn. A non-differentiable renderer takes this program and draws it on the current canvas. Consequently, a raster image is produced stroke-by-stroke. We first train this agent for two tasks by formulating appropriate loss functions: (i) unconditional character generation and (ii) parsing.

Unconditional character generation is the task of generating a novel concept[1] (i.e. character) given a dataset of concepts. For this task, our loss function includes the following components: an adversarial loss produced by a discriminator to make generated characters as "real" as possible, and two data fidelity losses assessing the conformity of the current canvas with the statistical properties of the overall dataset. We also use an additional entropy loss to prevent mode collapse.

In the parsing task, the goal for our agent is to reconstruct a given character (in raster-image) by drawing it through strokes using as few of them as possible. We utilize the same action space and environment as in the unconditional generation model, only difference being the input fed to the policy is a complete canvas to be reconstructed. Our reward function in this task has two components: a fidelity reward that indicates how much of a stroke is consistent with the target image and a penalty that increases with every 'continue' action being taken. This model explicitly learns the vectorization of the input raster-image in a self-supervised manner.

Next, we show that our parsing model can be exploited for exemplar generation (i.e. a novel drawing of a given character) and novel concept generation from type (i.e. novel character generation given an alphabet of 10 characters) *without any further training*. Given a character, the policy network of our parsing model outputs a distribution over the action space where likelihood of actions at each time step eventually allows us to generate variations of the input image. For novel concept generation conditioned on a type (i.e. alphabet), we compose a stroke library by parsing the provided inputs. As we sample strokes from this library, we observe novel samples forming, in coherence with the overall structure of the alphabet. *To the best of our knowledge, we are the first to tackle these tasks with a self-supervised approach that operates on stroke space.*

Through experiments we show that our agent can successfully generate novel characters in all three ways (unconditionally, conditioned on a given alphabet, conditioned on a given character), and parse and reconstruct input characters. For both exemplar generation and type conditioned novel concept generation, we provide LPIPS (Zhang et al., 2018), L2 and SSIM measures between input samples and generated images.

Our contributions in this paper are two-fold: (i) we present a drawing agent that can successfully handle all of the generation and parsing tasks in the Omniglot challenge in a self-supervised, stroke-

---

[1]Omniglot challenge terminology.

based manner – such a model did not exist (ii) we provide for the first time perceptual similarity based quantitative benchmarks for the 'exemplar generation' and 'type conditioned novel concept generation' tasks.

## 2    RELATED WORK

The main purpose of this work is to present a self-supervised approach in order to solve the generation and parsing tasks in the Omniglot Challenge (Lake et al., 2015), by capturing the stroke-level representation of images. Here we initially examine the supervised and self-supervised approaches to Omniglot challenge. Then, we review the work on image vectorization. And lastly, we touch upon the research on program synthesis in the context of this study.

**Omniglot Challenge**    Omniglot dataset of world alphabets was released with a set of challenges: parsing a given letter, one shot classification, generating a new letter given an alphabet, generating a novel sample of a character, and unconditional generation. Omniglot letters have samples that are conditionally independent based on the alphabet-character hierarchy, hence, a distinctive approach to achieve all these tasks is Hierarchical Bayesian modeling (Lake et al., 2015), (Lake et al., 2013). As the Omniglot letters included human strokes as labels, the compositional and causal nature of letters are leveraged to model the generation process. Later, neurosymbolic models are also shown to be successful for unconditional generation (Feinman & Lake, 2020a) and conceptual compression for multiple tasks presented within the Omniglot Challenge (Feinman & Lake, 2020b).

However, without the stroke set that generated a concept, these tasks become more difficult. The idea of sequential image generation is examined by recurrent VAE models (Rezende et al., 2016), (Gregor et al., 2015), (Gregor et al., 2016). DRAW (Gregor et al., 2015) and Convolutional DRAW (Gregor et al., 2016) were able to generate quality unconditional samples from MNIST and Omniglot datasets respectively. DRAW is proposed as an algorithm to generate images recurrently. The network is able to iteratively generate a given image by attending to certain parts of the input at each time step. Convolutional DRAW improved the idea with an RNN/VAE based algorithm that can capture the global structure and low-level details of an image separately in order to increase the quality of generations. Later, it is shown that Hierarchical Bayesian Modeling can be improved by the representational power of deep learning and attentional mechanisms in order to achieve three of the five Omniglot challenges (Rezende et al., 2016). Another novel idea to leverage Bayesian modeling to tackle Omniglot Challenge was performing modifications on the VAE architecture to represent hierarchical datasets (Edwards & Storkey, 2016) (Hewitt et al., 2018). The significance of these studies is that they were able obtain latent variables to describe class-level features effectively. Despite the ability to utilize the same model for different problems (one-shot classification, unconditional and conditional generation), raster-based one-step generative models have two disadvantages we want to address. First, they cannot leverage the higher level abstraction and quality comes with working on a vector space. Secondly, one-step generation does not provide an interpretable compositional and causal process describing how a character is generated. In this work, we combine the advantages of two groups of aforementioned models with an agent operating on stroke representation of images that uses only raster images during training. Thus, we aim to solve all three generative and the parsing (reconstruction) tasks of the Omniglot challenge. We show that the model trained for reconstruction can also be adopted as a tool that captures the compositional structure of a given character. Without any further training, our agent can solve exemplar generation and type conditioned novel concept generation problems.

**Image Generation by Vectorization — With Stroke Supervision**    Sketch-RNN (Ha & Eck, 2017) is the first LSTM/VAE based sketch generation algorithm. It is later improved to generate multiclass samples (Cao et al., 2019) and increase the quality of generations by representing strokes as Bezier curves (Song, 2020). The idea of obtaining a generalizable latent space by image-stroke mapping is studied by many (Aksan et al., 2020; Das et al., 2021; Bhunia et al., 2021; Wang et al., 2020). In CoSE (Aksan et al., 2020), the problem is articulated as 'completion of partially drawn sketch'. They achieved state of the art reconstruction performance by utilizing variable-length strokes and a novel relational model that is able to capture the global structure of the sketch. The progress in stroke representation is continued with incorporation of variable-degree Bezier curves (Das et al., 2021), and capturing Gestalt structure of partially occluded sketches (Lin et al., 2020).

**Self Supervised Vectorization** Self-supervised vector-based image generation problem has been approached by RL based frameworks (Zhou et al., 2018), (Ganin et al., 2018), (Mellor et al., 2019), (Huang et al., 2019), (Schaldenbrand & Oh, 2020), and (Zou et al., 2020). In SPIRAL (Ganin et al., 2018), unconditional generation and reconstruction tasks are tackled with adversarially trained RL agents. Succeeding research enhanced the reconstruction process by a differentiable renderer, making it possible for agents to operate on a continuous space (Huang et al., 2019; Schaldenbrand & Oh, 2020). In order to avert the computational expense of RL based algorithms, end-to-end differentiable models are developed through altering the rendering process (Nakano, 2019) or formulating the generation process as a parameter search (Zou et al., 2020). More recently, a differentiable renderer and compositor is utilized for generating closed Bezier paths and the final image respectively (Reddy et al., 2021). This method led to successful interpolation, reconstruction, and sampling processes. Most related to our work is SPIRAL where both reconstruction and unconditional generation is studied through self-supervised deep reinforcement learning. However, our approach has some significant differences. First, in SPIRAL each stroke is also represented as a Bezier curve, yet, the starting point of each curve is set as the final point of the previous curve. In our model, all control points of the Bezier curve are predicted by the agent at each time step. Hence, the agent has to learn the continuity and the compositionality of the given character in order to produce quality samples. Secondly, SPIRAL provides a generative model that works through a graphics renderer without addressing the conditional generation problem. They show impressive results on both natural images and handwritten characters. While we provide a solution for multiple generative tasks, we have not explored our model in the context of natural images. Another approach that presents a similar scheme to the reconstruction problem is "Learning to Paint" (Huang et al., 2019). In Learning to Paint, the proposed model is utilized specifically for reconstruction. When reconstruction is considered, the main difference of our model is that since we try to model a human-like generation process, our agent outputs a single stroke at each time step with the environment being altered throughout this process while in Learning to Paint, 5 strokes are predicted by the agent at each time step. As a major difference from previous studies, our agent decides whether to stop or keep drawing before generating a stroke. This enables the agent to synthesize an image with as few actions as possible when motivated with our reward formulations.

**Self Supervised Program Synthesis** Our method essentially outputs a visual program that depends only on the rastered data. In that sense, studies on Constructive Solid Geometry (CSG)are also related. Different RL frameworks for reconstruction of a given CSG image, that is essentially a composition of geometric shapes, are proposed (Ellis et al., 2019; Zhou et al., 2020). The former considered parsing as a search problem that is solved by using a read-eval-print-loop within a Markov Decision Process. The latter adopted a Tree-LSTM model to eliminate invalid programs and the reward is considered to be the Chamfer distance between the target image and current canvas.

## 3 METHOD

Our model consists of a policy network and a (non-differentiable) renderer. At time step $t$, the policy network takes the current canvas, $C_t$ – a raster-image, as input and outputs two distributions, $\pi_B$ and $\pi_S$. The first distribution, $\pi_B$, is for stroke (i.e. Bezier curve)-parameters and the second

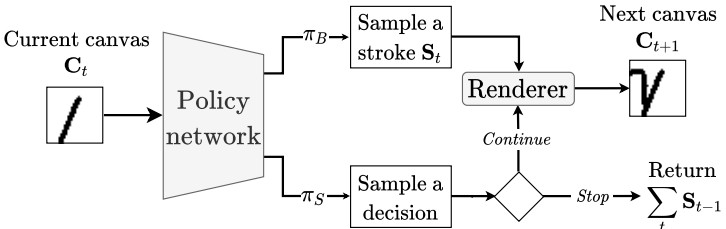

Figure 2: Generator model. At each time step, the policy network receives a canvas and outputs two distributions for Bezier curve parameters and stop/continue decision. When the 'continue' decision is sampled, the resulting stroke is rendered and added to the final output.

one, $\pi_S$, is for the continue/stop decision. From the first distribution, we randomly sample a stroke defined by its 7 parameters (x-y coordinates of start, end, control points of the quadratic Bezier curve, and a brush-width). From the second distribution, we randomly sample a decision. If the decision happens to be 'continue', we add the newly sampled stroke to the current canvas, $C_t$, increment time (i.e. $t \leftarrow t + 1$) and restart. If the decision was to 'stop', then $C_t$ is returned as the final output. Our model is able to handle parsing and different generation tasks, and the processing

pipeline we just described is common in all these tasks. What changes among tasks is the reward functions and/or training procedures, which we explain below.

**Unconditional Generation** The task of 'generating new concepts' as dubbed in Omniglot challenge, is essentially unconditional sampling from a distribution obtained from the whole Omniglot training set. Here, the model is asked to generate completely novel samples (i.e. characters) without any constraints. For this task, at each time step $t$, we calculate an instantaneous reward, $r_t$, that has three components:

$$r_t = D(\boldsymbol{C}_t) + \lambda_1 \mathrm{align}(\boldsymbol{C}_t, \boldsymbol{I}) + \lambda_2 \mathcal{N}(|\boldsymbol{C}_t|; \mu, \sigma). \tag{1}$$

The first term is a reward based on a discriminator to make generated characters as 'real' as possible. $D(\cdot)$ is a discriminator that outputs the "realness" score of its input canvas. We train it in an adversarial manner by using the generated examples as negatives and the elements of the input dataset as positives. The second term is a clustering-based data fidelity reward. The function $\mathrm{align}(\boldsymbol{C}_t, \mathbf{I})$ measures the alignment between the current canvas $\boldsymbol{C}_t$ and another canvas $\mathbf{I}$, which is a randomly selected cluster center at the beginning of each episode. The cluster centers are obtained by applying $k$-means on all characters in the input dataset. align basically counts the number of intersecting on-pixels (between the two canvases) minus the number of non-intersecting on-pixels in $\boldsymbol{C}_t$, and divides this quantity by the number of on-pixels in $\mathbf{I}$. The final term assesses the conformity of the current canvas with the dataset in terms of the number of on-pixels. $\mathcal{N}(|\boldsymbol{C}_t|; \mu, \sigma)$ evaluates a normal distribution with $(\mu, \sigma)$ at $|\boldsymbol{C}_t|$ which is the number of on-pixels in the current canvas. We obtain $(\mu, \sigma)$ by fitting a normal distribution to the on-pixel counts of characters in the training set. We observed that the second and third terms accelerate learning as they guide the exploration within the vicinity of real characters. During training, instead of using the instantaneous reward, $r_t$, we use the difference of successive rewards, i.e. $r_t - r_{t-1}$.

In order to encourage exploration and avoid mode collapse, we use an entropy penalty term as

$$\alpha \max(0, \mathrm{KL}([\pi_B, \pi_S], \mathrm{U}) - \tau). \tag{2}$$

Here, KL indicates KL-divergence and U is the uniform distribution. This term first measures the divergence between the uniform distribution and $\pi_B, \pi_S$, the distributions output by the policy network. Then, through the hinge function, if the divergence exceeds a threshold ($\tau$), this term activates and increases the penalty. The policy network and the discriminator $D$ are updated alternatingly after 256 images are generated at each iteration. We employ the REINFORCE algorithm (Williams, 1992) to update the weights of the policy network. Discriminator is trained using hinge loss. In order to stabilize the discriminator and keep the Lipschitz constant for the whole network equal to 1, Spectral Normalization is applied at each layer (Miyato et al., 2018). Throughout the training, we kept the balance ratio between generated and real samples at 3.

**Image Reconstruction by Parsing** In the "parsing" task, the goal is to reconstruct the given input image by re-drawing it through strokes as accurately as possible. To this end, we formulate a new reward function with two terms: a fidelity reward that indicates how much of a stroke is consistent with the input image (using the "align" function introduced above) and a penalty that is added with every time increment represented by $t$ as 'continue' decisions being made:

$$r_t = \mathrm{align}(\boldsymbol{S}_t, \boldsymbol{C}_t) - \lambda_1 t, \tag{3}$$

where $S_t$ is the newly sampled stroke and $C_t$ is the current canvas (input). Second term simply acts as a penalty for every 'continue' action. The first term ensures the sampled stroke to be well-aligned with the input and the second term forces the model to use as few strokes as possible. There is no need for a discriminator. This model explicitly learns the vectorization of the input raster-image in a self-supervised manner.

Apart from the different reward function, another crucial difference between the training of the unconditional generation model and the parsing model is how the input and output are handled. In unconditional generation, the newly-sampled stroke is added to the current canvas, whereas in

parsing, we do the opposite: the sampled stroke is removed (masked out) from the current canvas, and the returned final canvas is the combination of all sampled strokes until the 'stop' decision. $\lambda$, $\alpha$ and $\tau$ in Equations 1, 2, and 3 are hyperparameters adjusted experimentally. (see 'Training Details' in Appendix B ).

**Generating New Exemplars**   In this task, a model is required to generate a new exemplar (i.e. a variation) of an unseen concept (i.e. character). To the best of our knowledge, we are the first to tackle this task in a self-supervised stroke-based setting. Most importantly, we do not require any training to achieve this task. We utilize our parsing network described in the previous section to capture the overall structure of a given letter. In order to produce new exemplars, we randomly sample different parsings (a set of strokes) from the distribution generated by the agent. In order to eliminate 'unlikely' samples, we compute the likelihood of the parsing given the resulting policy, and apply a threshold.

**Generating Novel Concepts from Type**   In this task, the goal is to to generate a novel concept (i.e. character) given a previously unseen type (i.e. alphabet) consisting of 10 concepts. The novel concepts should conform to the overall structure, that is, the stroke formulation and composition of the given type (alphabet). We, again, tackle this challenge using our parsing network without any further training. To do so, we first parse all input images into its strokes. For each input image, we sample five stroke sets from the stroke-parameters distribution output by the policy network. During the sampling process, we again use the likelihood-based quality function described in the previous section. We add all the strokes sampled during this process to form a *stroke library*. Here the strokes are stored with the time steps they are generated. Noting that the number of strokes sampled for a given character is not constant, we approximate a distribution for stopping actions. This process provides a stroke set representing the structure of letters and the way they are composed, that is, we can exploit the compositionality and causality of an alphabet. Throughout the character generation process, a stroke is sampled at each time step belonging to that particular group of the library. The sampled strokes are summed together to obtain the final canvas.

## 4 EXPERIMENTS

**Datasets and Implementation Details**   We report generation and reconstruction (parsing) results on the Omniglot dataset (Lake et al., 2015), which includes 1623 characters from 50 different alphabets, with 20 samples for each character. 30 alphabets are used for training and the remaining 20 are used for evaluation. For unconditional generation and reconstruction, we also report results on the MNIST dataset (LeCun, 1998). For both datasets, we rescale input images to 32x32 in order for them to conform with our model.

Our policy network is composed of a ResNet feature extraction backbone and three MLP branches for computing the distributions over the action space. Architectural details can be found in Appendix A. For the Omniglot dataset, we take brush width as a constant and omit the corresponding MLP branch.

We tune the learning rate and weight decay of the generator, $\lambda$ hyperparameters in Equation 1 and Equation 3, $\alpha$ and $\tau$ hyperparameters in Equation 2, using the Tree-structured Parzen Estimator algorithm (Bergstra et al., 2011) in the RayTune library (Liaw et al., 2018).

For unconditional generation, we use the discriminator architecture proposed by Miyato et al. (2018). In order to stabilize the discriminator and keep the Lipschitz constant for the whole network equal to 1, Spectral Normalization is applied at each layer.  Discriminator is trained using the hinge loss. Throughout the training, we set the balance ratio between fake and real samples as 3. We performed hard-negative mining to speed up convergence during this process.

|  | FID |
|---|---|
| LSGAN (Mao et al., 2017) | $9.8 \pm 0.9$ |
| NSGAN (Goodfellow et al., 2014) | $6.8 \pm 0.5$ |
| WGAN (Arjovsky et al., 2017) | $6.7 \pm 0.4$ |
| WGAN GP (Gulrajani et al., 2017) | $20.3 \pm 5.0$ |
| DRAGAN (Kodali et al., 2017) | $7.6 \pm 0.4$ |
| VAE | $23.8 \pm 0.6$ |
| Ours | $17.3 \pm 3.0$ |

Table 1: Comparison of the FID scores for different models on the MNIST dataset. We report the mean and the variance of FID scores from 5 simulations with different weight initializations.

Figure 3: Quality of generated MNIST characters as training progresses (i.e. policy is updated) from left to right.

Figure 4: Omniglot unconditional samples. For randomly sampled generations, four closest samples (in terms of pixelwise L2 distance) from the training dataset are presented.

## 4.1 UNCONDITIONAL GENERATION

We initially tested our approach on the MNIST dataset. Figure 3 presents the improvement in the quality of samples generated throughout the policy network updates. At the beginning, generated characters are mostly random scribbles. Towards the end, they start to look like real digits. Table 1 shows that our method achieves an acceptable FID score (Heusel et al., 2017) given the scores of other prominent GAN and VAE methods. Presented FID values are taken from Lucic et al. (2017).

Figure 4 shows sample generations for the Omniglot dataset. To demonstrate that our generations are not duplicates of the characters in the training set, we present the four most similar characters from the training set to our generations. Similarity is computed using pixelwise L2 distance. Finally, Figure 5 presents more generated characters, which demonstrate the variability and the quality of generated concepts. The agent was able to capture the type of strokes, number of strokes a character has and letter structures without any stroke supervision.

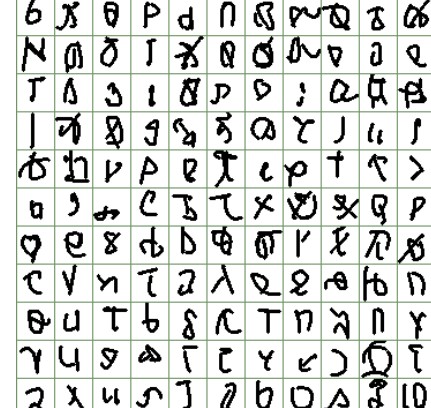

## 4.2 IMAGE RECONSTRUCTION BY PARSING

Figure 6 presents sample parsing and reconstruction results on MNIST. Our agent can reconstruct a character from the test set in a minimal number of actions within the abilities of quadratic Bezier curves. Selected brush widths also conform with the stroke heterogeneity of the dataset.

Figure 5: Randomly sampled unconditional generations for the Omniglot dataset.

Then, we train our model with the characters in the Omniglot training set. For evaluation, we utilize the evaluation set with completely novel characters from unseen alphabets. Thereby, we can see that our agent has learned how to parse a given character. Due to the penalty term that increases with the number of strokes, there is a tradeoff for the agent to replicate a character exactly and replicate it in a small number of actions. This indirectly demotivates the agent from retouching the image with small strokes to minimize the difference to the

Figure 6: MNIST reconstructions. For each sample on the left hand side of the columns, parsing processes are demonstrated. Colors represent the order of the strokes. (*pink*: first stroke, *green*: second stroke, *blue*: third stroke)

Figure 7: Omniglot reconstruction. For each sample on the left hand side of the columns, resulting reconstructions are demonstrated.

target. Results in Figure 7 show that overall structure of the target images are preserved, however, small details are lacking in some of the examples. This reflects on the distance measures (Table 2).

## 4.3 GENERATING NEW EXEMPLARS

For this task, we use the evaluation set of the Omniglot dataset. For each character in the test set, we sample 500 different parses from the policy. In Figure 8, it can be observed that given an unseen letter from a novel alphabet, our agent can sample from the resulting distribution, and output quality variations. The major indications of variation are structures of the strokes, number of actions to generate a sample and the fine details of certain characters. We compare each produced character with its corresponding input image using LPIPS, SSIM and L2 distance values. The mean and standard deviation of these values for the whole evaluation set are $0.078 \pm 0.002$, $0.616 \pm 0.018$ and $0.08 \pm 0.016$, respectively. Results per alphabet can be found in Appendix C.3.

## 4.4 GENERATING NOVEL CONCEPTS FROM TYPE

In order to generate a concept that is likely to belong in a given alphabet, we again leverage our reconstruction model. Given 10 different characters of an unseen alphabet, we are able to generate novel images with similar structural features. Results presented in Figure 9 show that our algorithm can model the compositional pattern of an alphabet in stroke space. In order to obtain quantitative results, (e.g. LPIPS, L2 and SSIM), we produce 10000 images conditioned on each input set and randomly sample characters by utilizing the discriminator trained for the unconditional generation model, assuming it has learned what features of a given input imply a real character. We generate

| Method | MNIST | Omniglot |
|---|---|---|
| ImageVAE | 0.0033 | N/A |
| Im2Vec (Reddy et al., 2021) | 0.0036 | N/A |
| Learning to Paint (Huang et al., 2019) | 0.006 | N/A |
| SPIRAL (Training distance) (Ganin et al., 2018) | 0.01 | 0.02 |
| StrokeNet (Training distance) (Zheng et al., 2018) | 0.015 | 0.02 |
| Ours | 0.04 | 0.06 |

Table 2: Reconstruction quality. L2 distance between target and the reconstructed image. (ImageVAE is taken from Reddy et al. (2021), it indicates a purely raster-based autoencoder. )

Figure 8: New exemplar generation. Given an unseen character from a new alphabet (highlighted in red boxes), the model generated 9 exemplars.

Figure 9: Novel sample generation conditioned on a type. Given 10 characters from an alphabet, our model produced 20 new samples.

a sampling distribution according to the discriminator scores of generated samples and repeat the sampling process multiple times for each input to obtain a set of outputs to be considered. For a sample generated, we calculate performance metrics with respect to all characters in the input. In order to report final metrics presented in supplemental figures 12b and 12a, we consider the most similar input-output pairs. The mean and standard deviation of LPIPS, SSIM and L2 values for the whole evaluation set are $0.0801 \pm 0.003$, $0.502 \pm 0.068$ and $0.1263 \pm 0.00086$ respectively.

## 5    CONCLUSION

We proposed a self-supervised reinforcement learning approach for stroke based image generation. We trained our model for unconditional generation and parsing on handwritten character datasets by defining a single action space and environment. Through experiments, we showed that, given the whole training set, our agent is able to capture the overall distribution and generate quality novel samples for the challenging Omniglot dataset. Then, we trained our agent for the parsing task; given a raster image, the goal is to reconstruct it through as few strokes as possible. We demonstrated that the parsing agent can be utilized for generating exemplars of a concept and creating novel samples conditioned on a type, without any further training, only difference being how it is called among tasks. To the best of our knowledge, we are the first to tackle these tasks with a self-supervised approach that operates on a stroke level. In this work, we used quadratic Bezier curves as the smallest unit of sketching. However, for human-level generations, the stroke representations should be enhanced to capture more complex structures. We anticipate that this will improve the overall performance.

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

## A  Network Architecture

The backbone is a ResNet with 3 convolutional layers and 8 residual layers. The first convolutional layer has 32 filters of size 5x5. The second and third convolutional layers have 32 filters of size 4x4 and stride of 2, resulting in a tensor with dimensions 8x8x32. Then, we use standard residual layers described in (He et al., 2016). Each convolutional layer is followed by a Batch Normalization process and ReLU activation. The output of the final residual layer is flattened to a 2048x1 vector to be processed by the MLPs. The first MLP outputs a set of distributions for each control point of the Bezier curve. It has 1 fully connected layer that outputs a 192x1 vector. This vector is reshaped to a 32x6 matrix where each 32x1 vector defines a distribution over the possible coordinates. The MLPs used for selecting the brush width and sampling the stop/continue decision consist of 2 layers with 64 and 2 neurons.

## B  Training Details

The hyperparamaters used for unconditional generation and reconstruction are presented in Table 3 and 4, respectively.

| | |
|---|---|
| $\lambda_1$ | 1.016 |
| $\lambda_2$ | 1 |
| $\alpha$ | 0.336 |
| $\tau$ | 0.415 |
| Policy network optimizer | AdamW |
| Policy network learning rate | $3.096e-05$ |
| Policy network weight decay | 0.0064 |
| Discriminator learning rate | 0.0001 |
| Batch size | 256 |

Table 3: Hyperparameters for unconditional generation. $\lambda_1$ and $\lambda_2$ refer to the respective hyperparameters in Equation 1. $\alpha$ and $tau$ refer to the respective hyperparameters of entropy penalty in Equation 2.

| | |
|---|---|
| $\lambda_1$ | 0.089 |
| $\alpha$ | 0.59 |
| $\tau$ | 2.72 |
| Policy network optimizer | AdamW |
| Policy network learning rate | $1.5e-4$ |
| Policy network weight decay | $1.6e-5$ |
| Batch size | 256 |

Table 4: Hyperparameters for reconstruction (parsing). $\lambda_1$ refers to the 'number of action' penalty in Equation 3. $\alpha$ and $tau$ refer to the respective hyperparameters of entropy penalty in Equation 2.

## C  Experiments: Supplemental Figures

### C.1  Unconditional Generation

In Figure 10, we present the FID values for the generated images along the training on Omniglot dataset.

### C.2  Parsing

In Table 5, we present the mean number of strokes our agent used to parse the characters for each alphabet in the test set.

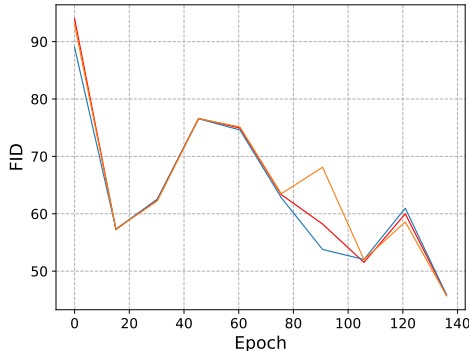

Figure 10: FID values for unconditional generations of Omniglot dataset throughout the training process. The experiment is repeated over 3 seeds.

| Alphabet | Number of Strokes | | Sample Image |
|---|---|---|---|
| | Our Model | Human Labeled Data | |
| Angelic | 3.935 | 4.49 | |
| Atemayar Qelisayer | 10.15 | 3.571 | |
| Atlantean | 6.209 | 2.078 | |
| Aurek-Besh | 7.6 | 2.565 | |
| Avesta | 9.511 | 1.52 | |
| Ge_ez | 10.112 | 1.984 | |
| Glagolitic | 5.24 | 2.88 | |
| Gurmukhi | 6.080 | 3.09 | |
| Kannada | 4.217 | 2.33 | |
| Keble | 8.573 | 4.140 | |
| Malayalam | 7.215 | 1.453 | |
| Manipuri | 10.676 | 2.82 | |
| Mongolian | 8.93 | 2.405 | |
| Old Church Slavonic | 5.171 | 2.954 | |
| Oriya | 5.59 | 2.82 | |
| Sylheti | 11.38 | 2.84 | |
| Syriac | 6.35 | 2.206 | |
| Tengwar | 8.088 | 2.492 | |
| Tibetan | 11.69 | 3.62 | |
| ULOG | 6.417 | 3.253 | |

Table 5: For each alphabet in the Omniglot evaluation set, we present the number of strokes our agent used to reconstruct the given image vs. mean number of strokes obtained from human-labeled data. The stroke count for human-labeled data is calculated using the labels within the Omniglot dataset.

## C.3 EXEMPLAR GENERATION

In Figure 11a, we demonstrate LPIPS metrics calculated by using 3 different backbones (AlexNet, VGG, and SqueezeNet). In Figure 11b, we present L2 and SSIM values. These metrics are calculated over all examples generated for the test set.

## C.4 GENERATING NOVEL CONCEPTS FROM TYPE

In Figure 12a, we demonstrate LPIPS metrics calculated by using 3 different backbones (AlexNet, VGG, and SqueezeNet). In Figure 12b, we present L2 and SSIM values.

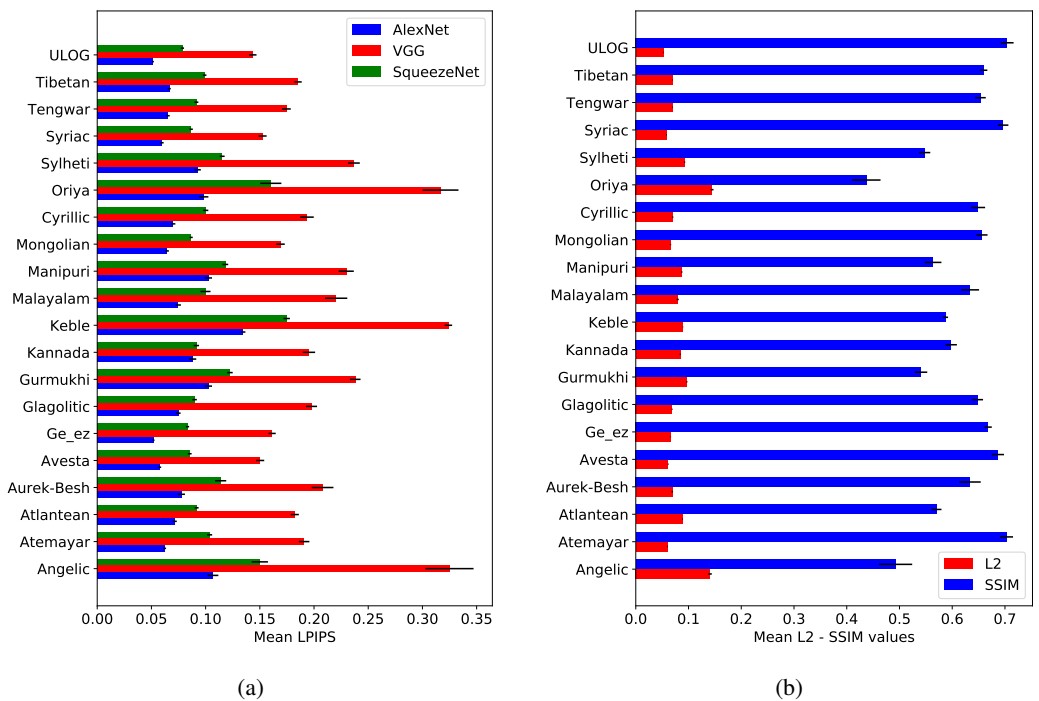

(a)

(b)

Figure 11: LPIPS values for each alphabet in the test set calculated from sampled exemplars (a), SSIM and L2 values for each alphabet in the test set calculated from sampled exemplars (b).

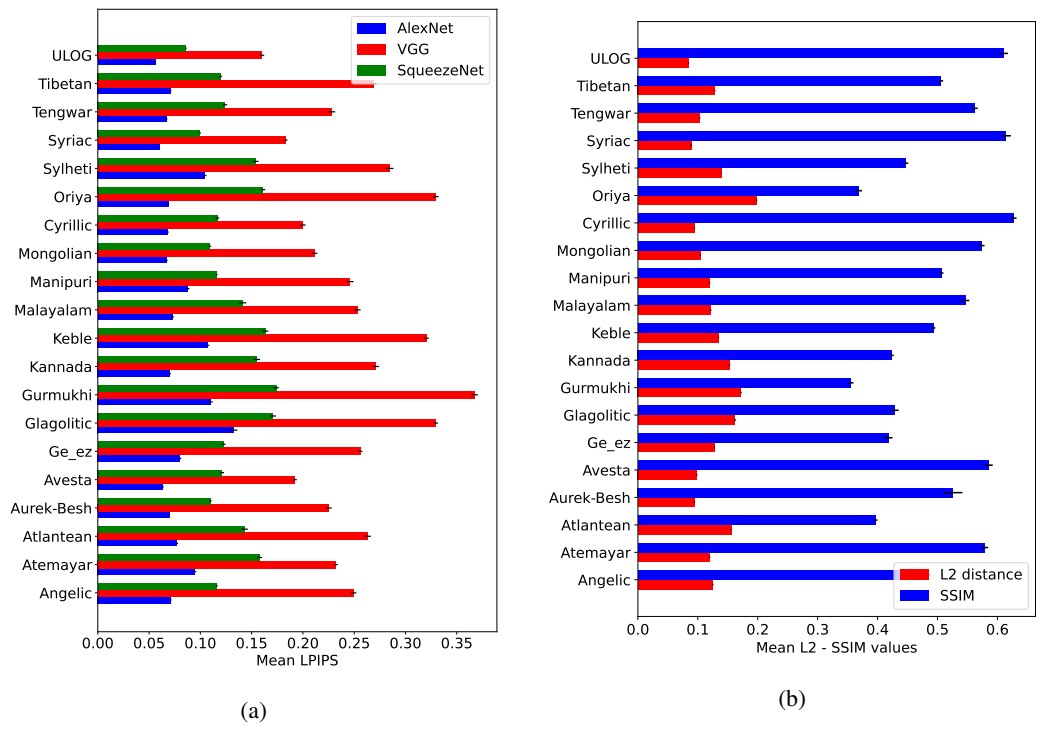

(a)

(b)

Figure 12: LPIPS values for each alphabet in the test set calculated from novel samples produced (a), L2-SSIM values for each alphabet in the test set calculated from novel samples produced (b).

