# OpenReview forum: "Character Generation through Self-Supervised Vectorization"
_ICLR.cc/2022/Conference — ICLR 2022 Submitted_

### Official Review · Reviewer_c9oL · 2021-11-02

**Correctness:** 2
**Technical Novelty And Significance:** 2
**Empirical Novelty And Significance:** 1
**Recommendation:** 3
**Confidence:** 4

**Main Review:**

Positives
---------

The overall approach proposed by the paper seems sound, and the results look visually reasonable in most cases.


Concerns
--------

My biggest concern is that the approach and results do not feel well situated in the context of advancements in self supervised vectorisation over the last 11 or so months. For example, these papers both propose differentiable approaches to drawing stroked curves (without the requirement that they be closed):

- Tzu-Mao Li, Michal Lukáč, Michaël Gharbi, and Jonathan Ragan-Kelley. 2020. Differentiable vector graphics rasterization for editing and learning. ACM Trans. Graph. 39, 6, Article 193 (December 2020), 15 pages. DOI:https://doi.org/10.1145/3414685.3417871
- Daniela Mihai, and Jonathon Hare. 2021. Differentiable Drawing and Sketching, arXiv preprint arXiv:2103.16194, https://arxiv.org/abs/2103.16194

In both those papers experiments were performed with character reconstruction, and the former demonstrates vector generation with a VAE. In both papers, like this one, the training of the models is self-supervised utilising only the rasters. Whilst I acknowledge that this paper explores more of the OmniGlot tasks, it's not obvious to me why the approaches in these other papers could not do the same thing (and certainly could be compared for unconditional parsing/reconstruction tasks).

My secondary concern would be that whilst many of the qualitative results look okay, many of the quantitative results using the proposed model seem not to be so good; for example in terms of reconstruction quality in table 2, the proposed model is significantly worse performing than all the compared approaches.

As a side note, it's not obvious to me why one would necessarily use perceptual models like SSIM or LPIPS for measuring performance in this particular task. I'd at least like to see some kind of justification for that choice.

**Summary Of The Paper:**

This paper presents an approach using reinforcement learning to parse and generate characters. Experiments are performed both with the omniglot challenge and MNIST datasets. In the context of omniglot, in addition to unconditional generation and parsing, results are also shown on the exemplar generation and type conditioned generation tasks. The proposed approach only uses pixel/raster level information during training and does not use stroke or vector data.

**Summary Of The Review:**

Overall, I think there are potentially some interesting ideas within this work, but there is a significant lack of context with respect to recent works in this space that would need to be addressed. Further, the results suggest in some cases the proposed methodology performs significantly worse than other approaches.

---

### Official Review · Reviewer_hg9W · 2021-11-03

**Correctness:** 3
**Technical Novelty And Significance:** 2
**Empirical Novelty And Significance:** 1
**Recommendation:** 5
**Confidence:** 3

**Main Review:**

In general, I think this submission is OK due to the following points:

1. The problem that this submission aims to solve, how to introduce high-level image information without the data limitation, is important.
2. The proposed method is simple and easy to implement.
3. The writting is good and I can follow it easily.

However, I have the following concerns, mainly about the experiment part:

1. I can understand that using the stroke information in the neural network is better than the pixel-based alternatives but the authors do not mark this advantages empirically. For example, as for me, I think there should be some experiments about the comparisons between the pixel-based methods and the stroke-based methods (at least the proposed one).
2. Incorporating higher level abstration of image information into the deep framework is widely studied, which is also mentioend by the authros in the related work part. So I wonder what's the improvement between the developed method and those methods (like DRAW). Specifically, I am really curious about the difference when comparing with DRAW, which also uses one single image as training data. Additionally, the paper states that DRAW cannot provide an interpretable compositional and causal process describing how a character is generated, which is not true.
3. I am confused about the functionality of each term in Eq. (1). Can the authors conduct some ablation studies to support the role of the last two terms, like why using the align operation can accelerate the training stage.
4. In terms of the baseline comparisons, the chosen algorithms are old (all before 2018). Even though, the proposed method cannot enjoy a quantitative evaluation merit in Tab. 1-2.
5. After reading the submission, I find the technical contribution minor. Except the two terms for speeding up the training, I cannot see any specific strategy (insights) to tackle the self-supervised image generation problem. Can the authors explain a bit here?

**Summary Of The Paper:**

In this paper, the authors present a method for character generation using the self-supervised technology. Different from existing approaches, it can leverage the benefits from higher level abstration (due to the used strokes) and get rid of stroke supervision. In this way, high-quality images are generated while the supervised training data requirements are relieved. Although some comparisons are made on Omniglot dataset, the advantages of this method are not very clear.

**Summary Of The Review:**

Generally speaking, I am not an expert in the field of stroke-based image generation and remain lukewarm about this project. Due to the strengths and weakness I mentioned above, I rate this submission as weak reject.

---

### Official Review · Reviewer_aFpU · 2021-11-03

**Correctness:** 3
**Technical Novelty And Significance:** 3
**Empirical Novelty And Significance:** 2
**Recommendation:** 5
**Confidence:** 4

**Main Review:**

[Paper weakness]
Modeling:
- The rewards are designed based on a discriminator. As we know, generative adversarial networks are not easy to train since generative networks and discriminative networks are trained alternatively. In the proposed method, the policy network and the discriminator are trained alternatively. I doubt if it is easy to train the model. I would like to see the training curves for rewards value.
-	The detailed alignment function used in Eq. (1) and Eq. (3) need to be provided.


Experiment:
-	The results are not satisfying. In the experiment, the generation quality of the proposed method is not good as traditional generative networks in terms of FID. In the image parsing part, the results are far behind the compared methods.
-	Since the results are not comparable to the existing methods, there seems not too much significance for the proposed methods.


**Summary Of The Paper:**

The paper proposes a self-supervised reinforcement learning approach to train a drawing agent for character generation and parsing. The drawing agent operates on the stroke-level (i.e. vector) representation of images. Different from the one-step generative model, the proposed method can capture the compositional structure of a given character. It is interesting to tackle these tasks with a self-supervised approach and reinforcement learning that operates on stroke space.


**Summary Of The Review:**

Overall, the paper proposes a self-supervised character generation and parsing by reinforcement learning. However, there are still some issues on modeling and training details that need to be clarified. Also, since the results are not comparable to the existing methods, it seems not significant for the proposed methods.

---

### Decision · Program_Chairs · 2022-01-20

**Decision:**

Reject

**Comment:**

The paper studies the problem of character generation using reinforcement learning for generation/parsing. All the reviewers recommended reject due to insufficient experimental investigation to support the ideas. The authors did not provide a rebuttal. Hence, the reviewers' opinion still remains the same. AC agrees with the reviewers and believes that the paper is not yet ready for publication.